# Prevalence and incidence of young onset dementia and associations with comorbidities: A study of data from the French national health data system

Laure Carcaillon-Bentata[1]*, Cécile Quintin[1], Marjorie Boussac-Zarebska[1], Alexis Elbaz[1,2]

1 Santé publique France, Saint-Maurice, France, 2 Université Paris-Saclay, Univ. Paris-Sud, UVSQ, CESP, Inserm, Villejuif, France

* laure.carcaillon-bentata@santepubliquefrance.fr

## Abstract

### Background

Dementia onset in those aged <65 years (young onset dementia, YOD) has dramatic individual and societal consequences. In the context of population aging, data on YOD are of major importance to anticipate needs for planning and allocation of health and social resources. Few studies have provided precise frequency estimates of YOD. The aim of this study is to provide YOD prevalence and incidence estimates in France and to study the contribution of comorbidities to YOD incidence.

### Methods and findings

Using data from the French national health data system (Système National des Données de Santé, SNDS) for 76% of the French population aged 40 to 64 years in 2016 ($n = 16,665,795$), we identified all persons with dementia based on at least 1 of 3 criteria: anti-Alzheimer drugs claims, hospitalization with the International Classification of Diseases-10th Revision (ICD-10) dementia codes (F00 to F03, G30, G31.0, G31.1, or F05.1), or registration for free healthcare for dementia. We estimated prevalence rate (PR) and incidence rate (IR) and estimated the association of comorbidities with incident YOD. Sex differences were investigated. We identified 18,466 ($PR_{standardized} = 109.7/100,000$) and 4,074 incident ($IR_{standardized} = 24.4/100,000$ person-years) persons with prevalent and incident YOD, respectively. PR and IR sharply increased with age. Age-adjusted PR and IR were 33% (95% confidence interval (CI) = 29 to 37) and 39% (95% CI = 31 to 48) higher in men than women ($p < 0.001$ both for PR and IR). Cardio- and cerebrovascular, neurological, psychiatric diseases, and traumatic brain injury prevalence were associated with incident YOD (age- and sex-adjusted $p$-values <0.001 for all comorbidities examined, except $p = 0.109$ for antihypertensive drug therapy). Adjustment for all comorbidities explained more than 55% of the sex difference in YOD incidence. The lack of information regarding dementia subtypes is the main limitation of this study.

**Data Availability Statement:** Data cannot be shared as access to the SNDS has to be approved by the French Data Protection Authority. Interested

investigators can request their own access to the data according to the French law. Information on obtaining access to SNDS data can be found at https://www.snds.gouv.fr/SNDS/Processus-d-acces-aux-donnees. Requests for access to SNDS data must be submitted to the National Institute for Health Data (INDS) (https://www.health-data-hub.fr/demarches-reglementaires).

**Funding:** The author(s) received no specific funding for this work.

**Competing interests:** The authors have declared that no competing interests exist.

**Abbreviations:** ATC, anatomical therapeutic chemical classification; CI, confidence interval; CNAMTS, Caisse Nationale d'Assurance Maladie des Travailleurs Salariés; ICD-10, International Classification of Diseases-10th Revision; IR, incidence rate; LOD, late onset dementia; LTD, long-term chronic disease; MMSE, Mini-Mental State Examination; OR, odds ratio; PR, prevalence rate; RNIAM, Répertoire National Interrégimes des Bénéficiaires de l'Assurance Maladie; SD, standard deviation; SNDS, Système National des Données de Santé; STROBE, Strengthening the Reporting of Observational Studies in Epidemiology; YOD, young onset dementia.

## Conclusions

We estimated that there were approximately 24,000 and approximately 5,300 persons with prevalent and incident YOD, respectively, in France in 2016. The higher YOD frequency in men may be partly explained by higher prevalence of cardiovascular and neurovascular diseases, substance abuse disorders, and traumatic brain injury and warrants further investigation.

## Author summary

### Why was this study done?

- Dementia onset in people aged <65 years (young onset dementia, YOD) has dramatic individual and societal consequences.

- To the best of our knowledge, there is a lack of data regarding the frequency of YOD in France.

- Few data are available regarding the contribution of comorbidities to YOD incidence.

### What did the researchers do and find?

- Using data from the French national health data system for 76% of the French population, we estimated YOD prevalence and incidence in individuals aged 40 to 64 years and examined the contribution of comorbidities to YOD incidence.

- Standardized prevalence rate (PR) and incidence rate (IR) were 109.7/100,000 and 24.4/100,000 person-years, respectively, and were higher in men than women.

- Cardiovascular and neurovascular, neurological, psychiatric diseases, and traumatic brain injury explained >55% of the sex difference in YOD incidence.

### What do these findings mean?

- We extrapolate that there were approximately 24,000 and approximately 5,300 persons with prevalent and incident YOD, respectively, in France in 2016.

- YOD incidence is higher in men than women, and this could possibly be due, at least in part, to the higher prevalence of cardiovascular and neurovascular diseases and substance abuse disorders in men.

## Introduction

Dementia represents a major worldwide public health concern as the number of people affected doubled between 1990 and 2016 and will continue to rise due to population aging [1]. Although dementia mainly occurs during old age, it can develop earlier. Young onset dementia

(YOD), also referred to as early onset dementia, generally refers to dementia occurring before 65 years [2]. Among people older than 60 years, 6% to 9% of all dementia cases occurred before 65 years [3].

Compared with individuals with late onset dementia (LOD), those suffering from YOD usually present more atypical symptoms; memory loss complaints are not systematically reported, while behavioral problems (e.g., mood disturbances and psychosis) often coexist [4]. These individuals face major psychological, familial, and socioeconomic challenges, while often still active professionally [5]. YOD has a major impact on the quality of life of people with the disease and their caregivers [6,7]. YOD burden is high for the health and social care systems in charge of young individuals affected by a chronic disease requiring a high and costly level of care.

In the context of population aging, data on YOD are of major importance to anticipate needs for planning and allocation of health and social resources [8]. Some field and register studies provided heterogeneous estimates of YOD prevalence, ranging from 38 to 420/100,000 between 30 and 64 years [9–17]. Fewer studies are available regarding YOD incidence, and methodological differences likely explain inconsistent results between them [9–11,16–19]. In France, to our knowledge, only 1 study reported on the prevalence of young onset Alzheimer disease and found it to be 41.2/100,000 before the age of 60 years in 1998 [20].

There is a growing interest in using administrative health databases to estimate the burden of chronic diseases, including dementia [21,22]. Given the high number of undiagnosed and untreated dementia in the population [23], these sources suffer from poor sensitivity [24]. However, their sensitivity may be higher in younger individuals [25,26]. In France, a preliminary study compared dementia prevalence based on administrative data to those from cohort studies and showed that administrative data underestimated prevalence after 70 years but yielded consistent estimates for YOD [27]. We hypothesized that the dramatic impact of YOD on patient's lives is more likely to lead younger individuals to be in contact with the health system than older ones.

In this paper, we used more recent data from the French national health data system for 76% of the French population to estimate YOD prevalence and incidence in 2016, overall and by age and sex, and to examine the contribution of comorbidities to YOD incidence.

## Methods

### Data sources

Data come from the French national health data system, Système National des Données de Santé (SNDS) [22]. It includes data on ambulatory healthcare consumption (including drug reimbursements and laboratory tests) and hospitalization records in all public/private hospitals since 2006. Drugs are coded using the anatomical therapeutic chemical classification (ATC). Data on hospitalizations in short-stay departments, rehabilitation, psychiatry, and at home are available. After each hospital stay, a discharge summary is filed, including a principal diagnosis (DP) as well as related (DR) and associated (DA) diagnoses coded using the International Classification of Diseases-10th Revision (ICD-10). In addition, in France, individuals with long-term chronic diseases (LTDs), including dementia, can benefit from free healthcare, and this information is registered in SNDS. Demographic information (age, sex, and vital status) is available.

SNDS gathers data from various social security schemes (which depend on occupation), covering >98% of the French population. Our analyses are restricted to the general scheme that includes persons employed in the private sector and spouses if unemployed (76% of the French population). As the SNDS was initially developed for the general scheme, data for its

affiliates are more exhaustive than for other schemes, especially for historical data. We excluded from our analyses workers in public services (11%), farmers, agricultural workers and their spouses if unemployed (5%), self-employed workers (6%), and workers from other smaller health systems (2%). As a governmental agency in charge of health surveillance in France, *Santé publique France* has a full access to SNDS data according to the French law to achieve its missions (French public health code, article R1461-14.1); the current project falls within the surveillance missions of *Santé publique France*.

### Identification of persons with YOD

We identified persons with dementia among all individuals aged 40 to 64 years old and alive on December 31, 2016. Dementia ascertainment relied on the presence of at least 1 of the following criteria: (1) ≥2 reimbursements over a 1-year period of anti-Alzheimer drugs (acetyl-cholinesterase inhibitors: Donepezil, ATC code N06DA02; Rivastigmine, N06DA03; Galantamine, N06DA04; Memantine, N06DX01); (2) LTD registration for dementia (ICD-10 codes: F00 to F03, G30, G31.0, G31.1, or F05.1); and (3) at least 1 hospitalization with a DP, DR, or DA diagnosis of dementia (same ICD-10 codes as above). To reduce the number of false positives and increase specificity, we excluded individuals who were identified only by one hospitalization with a DA or DR dementia diagnosis.

We previously showed that data collected over a 1-year period had poor sensitivity to identify prevalent dementia [28]. Consequently, in order to identify prevalent dementia in 2016 and 2015, the 3 criteria were searched over 5-year periods (2012 to 2016 and 2011 to 2015, respectively). Persons with incident dementia in 2016 were those identified in 2016 but not prevalent in 2015. We excluded persons with incident dementia without any healthcare use over 3 years before 2016, in whom we could not determine the exact incidence date.

### Comorbidities ascertainment

Based on the literature on YOD and LOD, we selected cardio- and cerebrovascular diseases and treatments, metabolic conditions, psychiatric diseases, psychotropic drugs, other neurological diseases, and traumatic brain injury as potential risk factors for YOD [29–35].

The identification of all comorbidities, except traumatic brain injury, relied on algorithms developed by SNDS experts in collaboration with the institution in charge of the SNDS, the *Caisse Nationale d'Assurance Maladie des Travailleurs Salariés* (CNAMTS) [36]. We identified traumatic brain injury using ICD-10 codes recommended by a literature review on ascertainment of neurotrauma in administrative databases [37]. A detailed description of the algorithms is available as S1 Table.

### Statistical analysis

Characteristics of individuals with prevalent and incident YOD overall and by sex are presented using means (standard deviation, SD) and number (proportions). Differences between sexes were tested using Student *t* test and Pearson chi-squared for continuous and categorical variables, respectively.

To estimate the YOD prevalence rate (PR) in 2016, we divided the number of individuals with prevalent dementia, aged 40 to 64 years old and alive on December 31, 2016, by the number of individuals affiliated to the general scheme of the social security at the same date provided by the French national health insurance register (Répertoire National Interrégimes des Bénéficiaires de l'Assurance Maladie, RNIAM). The YOD incidence rate (IR) in 2016 was estimated by dividing the number of individuals with incident dementia in 2016 by the number of

person-years affiliated to the general scheme in 2016, from which we subtracted the number of individuals with prevalent YOD in 2016.

PR and IR were computed overall and by sex and 5-year age groups. We used Poisson regression to estimate 95% confidence intervals (CI) and male-to-female prevalence ($PRR_{M/W}$) and incidence ($IRR_{M/W}$) rate ratios and 95% CI, overall (adjusted for age) and by age. We tested whether $PRR_{M/W}$ and $IRR_{M/W}$ changed with continuous age by including an interaction term between age and sex. We estimated age- and sex-standardized PR and IR using the French population as the reference (National Institute of Statistics and Economic Studies) [38].

The number of persons with prevalent and incident YOD in France in 2016 were extrapolated by applying age- and sex-specific PR and IR to the French population.

We compared the prevalence of comorbidities in incident YOD and the total population aged 40 to 64 years old affiliated to the general scheme in 2016; it comprised all persons without YOD who were reimbursed at least 1 healthcare intervention or hospitalized within 3 years before 2016. For each comorbidity, we computed odds ratios (ORs, 95% CI) using logistic regression adjusted for age and sex.

Analyses regarding prevalence, incidence, and associations of comorbidities with incident YOD were preplanned. As we found a strong difference in YOD incidence by sex, we further examined the contribution of comorbidities to sex differences in YOD incidence by estimating the percentage reduction in the association between sex and YOD after adjustment for comorbidities. Traumatic brain injury was included in our analyses after the main analyses were completed in response to a reviewer's suggestion.

This study is reported as per the Strengthening the Reporting of Observational Studies in Epidemiology (STROBE) guideline (S1 STROBE Checklist).

## Results

In 2016, we identified 18,466 persons with prevalent YOD (54.2% men, 45.8% women; Table 1). Their mean age was 57.3 (SD = 6.1) years. Hospitalization records and LTD registration for dementia were present in 56.3% and 53.4% of prevalent YOD, respectively. More women (32.2%) than men (23.5%) had anti-Alzheimer drug claims ($p < 0.0001$). Women (12.7%) more often met all 3 criteria for inclusion compared to men (7.9%) ($p < 0.0001$).

In 2016, 4,074 persons (55.4% men and 44.6% women) developed YOD. Their mean age was 57.2 (SD = 6.0) years, similar in men and women ($p = 0.065$). Hospitalization records and LTD registration for dementia were present in 53.5% and 49.2% of incident YOD, respectively. Anti-Alzheimer drug claims were less frequent in incident than prevalent YOD, but remained more frequent in women (21.4%) than men (15.6%) ($p < 0.0001$).

### PR and IR

Fig 1 and S2 and S3 Tables show PR and IR of YOD. Crude PR and IR were 105.4/100,000 persons and 23.4/100,000 person-years, respectively. Age- and sex-standardized PR and IR were 109.7/100,000 persons and 24.4/100,000 person-years, respectively. PR increased from 25.2/100,000 between 40 and 44 years to 284.8 between 60 and 64 years. Similarly, IR increased from 6.0/100,000 person-years in the younger age category to 62.3 in the older one.

By applying age/sex-specific rates to the French population, we estimate that approximately 24,000 persons (13,400 men and 10,600 women) aged 40 to 64 years had YOD in 2016 and that approximately 5,300 persons (3,000 men and 2,300 women) developed YOD in 2016.

Age-standardized PR and IR were higher in men ($PR_{men} = 125.5/100,000$; $IR_{men} = 28.5/100,000$ person-years) than women ($PR_{women} = 94.8/100,000$; $IR_{women} = 20.5/100,000$

**Table 1. Characteristics of prevalent and incident YOD, overall and by sex, in France (2016).**

| | Prevalent YOD | | | | Incident YOD | | | |
|---|---|---|---|---|---|---|---|---|
| | Overall | Men | Women | *p*-value | Overall | Men | Women | *p*-value |
| **Characteristics** | *n* = 18,466 | *n* = 10,011 | *n* = 8,455 | | *n* = 4,074 | *n* = 2,256 | *n* = 1,818 | |
| **Age, mean (SD)** | 57.3 (6.1) | 57.0 (6.3) | 57.6 (5.9) | <0.001 | 57.2 (6.0) | 57.0 (6.3) | 57.4 (6.0) | 0.065 |
| **Dementia criteria, *n* (%)** | | | | | | | | |
| Hospitalization record | 10,402 (56.3) | 5,578 (55.7) | 4,824 (57.1) | 0.404 | 2,180 (53.5) | 1,197 (53.1) | 983 (54.0) | 0.532 |
| Anti-Alzheimer drug claim | 5,072 (27.5) | 2,352 (23.5) | 2,720 (32.2) | <0.001 | 741 (18.2) | 351 (15.6) | 390 (21.4) | <0.001 |
| LTD registration for dementia | 9,863 (53.4) | 5,231 (52.3) | 4,632 (54.8) | 0.003 | 2,003 (49.2) | 1,116 (49.5) | 887 (48.8) | 0.675 |
| **Number of dementia criteria, *n* (%)** | | | | <0.001 | | | | <0.001 |
| One criteria alone | 13,549 (72.9) | 7,654 (76.5) | 5,805 (68.7) | | 3,372 (82.7) | 1,918 (85) | 1,453 (80.0) | |
| Two criteria | 3,143 (17.0) | 1,564 (15.6) | 1,579 (18.7) | | 556 (13.6) | 268 (11.9) | 288 (15.8) | |
| Three criteria | 1,864 (10.1) | 793 (7.9) | 1,071 (12.7) | | 147 (3.6) | 70 (3.1) | 77 (4.2) | |
| **Combination of dementia criteria, *n* (%)** | | | | <0.001 | | | | <0.001 |
| Hospitalization alone | 6,439 (34.9) | 3,669 (36.7) | 2,770 (32.8) | | 1,662 (40.8) | 944 (41.8) | 718 (39.5) | |
| Anti-Alzheimer drug claim alone | 1,521 (8.2) | 788 (7.9) | 733 (8.7) | | 316 (7.8) | 154 (6.8) | 162 (8.9) | |
| LTD registration alone | 5,499 (29.8) | 3,197 (31.9) | 2,302 (27.2) | | 1,394 (34.2) | 820 (36.3) | 573 (31.5) | |
| Hospitalization + anti-Alzheimer drug claim | 643 (3.5) | 323 (3.2) | 320 (3.8) | | 93 (2.3) | 42 (1.9) | 51 (2.8) | |
| Hospitalization + LTD registration | 1,456 (7.9) | 793 (7.9) | 663 (7.8) | | 278 (6.8) | 141 (6.3) | 137 (7.5) | |
| Anti-Alzheimer drug claim + LTD registration | 1,044 (5.7) | 448 (4.5) | 596 (7.1) | | 185 (4.5) | 85 (3.8) | 100 (5.5) | |

LTD, long-term chronic disease; SD, standard deviation; YOD, young onset dementia.

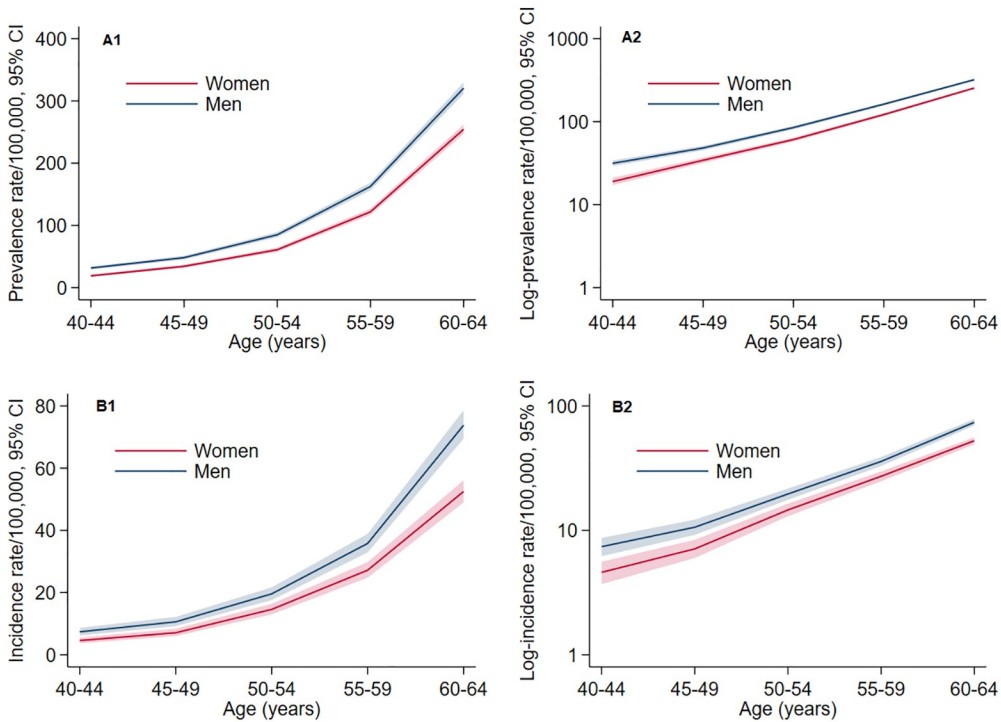

**Fig 1.** Age-specific prevalence (A1; A2, log scale) and incidence (B1; B2, log scale) rates of YOD in men and women (2016). Shaded regions represent 95% CIs. CI, confidence interval; YOD, young onset dementia.

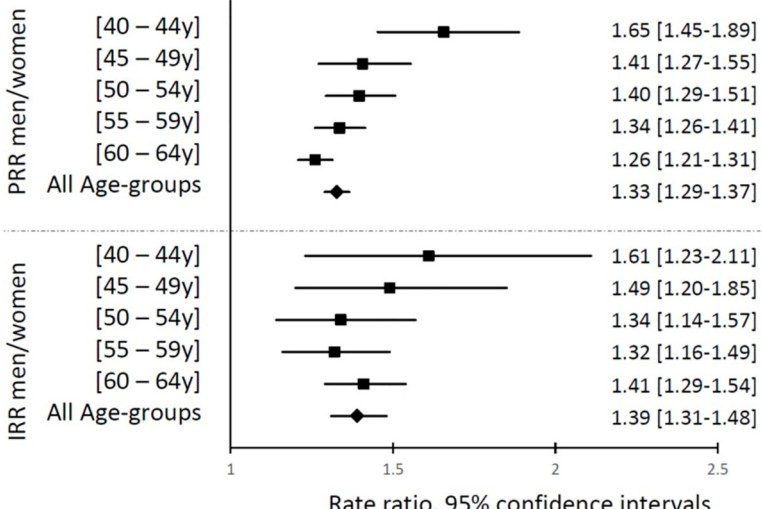

**Fig 2. Male-to-female PR (PRR_{M/W}) and IR (IRR_{M/W}) rate ratios, overall and by age.** Black bars represent 95% CIs. CI, confidence interval; IR, incidence rate; PR, prevalence rate.

person-years). The age-adjusted PRR_{M/W} was 1.33 (95% CI = 1.29 to 1.37), and the age-adjusted IRR_{M/W} was 1.39 (95% CI = 1.31 to 1.48). PRR_{M/W} decreased with age (Fig 2). It was 1.66 (95% CI = 1.45 to 1.89) between 40 and 44 years, and it progressively decreased to 1.26 (95% CI = 1.21 to 1.31) between 60 and 64 years (p-interaction between sex and continuous age <0.001). Age-specific IRR_{M/W} were similar to corresponding age-specific PRR_{M/W}, and there was a similar trend of decreasing IRR_{M/W} with continuous age, but it was not statistically significant (p-interaction = 0.494), possibly due to insufficient statistical power as the number of incident cases in each age group was considerably smaller than the number of prevalent cases. The attenuation of the difference between men and women with increasing age is due to a faster increase of PR and IR with age in women than men (Fig 1).

## Contribution of comorbidities to YOD

Table 2 shows a higher prevalence of all comorbidities in YOD than in the general population, in particular of cerebrovascular and neurological diseases, substance abuse, schizophrenia and psychotic disorders, and traumatic brain injury. In persons without YOD, all comorbidities were more frequent in men than women (S4 Table).

The age-adjusted OR of YOD for men compared to women was 1.48 (95% CI = 1.39 to 1.57, p < 0.001). Table 3 shows the contribution of comorbidities to this association; cardiovascular and neurovascular and metabolic diseases and substance abuse disorders explained 23.5% and 29.2% of the association, respectively. After adjustment for all comorbidities, the association decreased by 55.6% (OR = 1.19, 95% CI = 1.11 to 1.27, p < 0.001).

## Discussion

We used data from the French national health data system for 76% of the French population to estimate the PR and IR of YOD in people aged 40 to 64 years old. In 2016, age and sex-standardized PR and IR were 109.7/100,000 persons and 24.4/100,000 person-years, respectively. By applying these rates to the French population, we estimate that approximately 24,000 persons aged 40 to 64 years old had YOD in 2016, and approximately 5,300 persons developed

**Table 2. Prevalence of comorbidities in incident YOD and in persons without YOD from the French population (40 to 64 years) in 2016.**

| | Incident YOD (2016) | | Overall French population without YOD (2016)[a] | | Crude OR[b] | (95% CI) | p-value | Adjusted OR[b,c] | (95% CI) | p-value |
|---|---|---|---|---|---|---|---|---|---|---|
| | n | (%) | n | (%) | | | | | | |
| **Male sex** | 2,256 | (55.4) | 7,490,943 | (46.2) | 1.45 | (1.36 to 1.54) | <0.001 | 1.48 | (1.39 to 1.57) | <0.001 |
| **Cardiovascular, cerebrovascular, and metabolic diseases and drug-related therapies** | | | | | | | | | | |
| Morbid obesity | 368 | (9.0) | 729,979 | (4.5) | 2.11 | (1.09 to 2.35) | <0.001 | 1.75 | (1.57 to 1.95) | <0.001 |
| Diabetes | 549 | (13.5) | 1,068,192 | (6.6) | 2.21 | (2.02 to 2.42) | <0.001 | 1.36 | (1.24 to 1.49) | <0.001 |
| Acute cerebrovascular disease (excluding transient attacks) | 121 | (3.0) | 20,523 | (0.1) | 24.17 | (20.16 to 28.97) | <0.001 | 16.27 | (13.55 to 19.52) | <0.001 |
| Sequelae of cerebrovascular disease or history of acute cerebrovascular disease | 468 | (11.5) | 148,449 | (0.9) | 14.05 | (12.76 to 15.48) | <0.001 | 9.64 | (8.75 to 10.63) | <0.001 |
| Acute ischemic heart disease | 23 | (0.6) | 27,128 | (0.2) | 3.40 | (2.25 to 5.11) | <0.001 | 2.29 | (1.52 to 3.45) | <0.001 |
| Chronic ischemic heart disease or history of acute ischemic heart disease | 257 | (6.3) | 381,042 | (2.3) | 2.80 | (2.47 to 3.18) | <0.001 | 1.55 | (1.36 to 1.76) | <0.001 |
| Cardiac arrhythmias and conduction disorders | 220 | (5.4) | 181,704 | (1.1) | 5.04 | (4.40 to 5.77) | <0.001 | 3.00 | (2.61 to 3.44) | <0.001 |
| Antihypertensive drug therapy | 1,275 | (31.3) | 3,276,569 | (20.2) | 1.80 | (1.68 to 1.92) | <0.001 | 1.06 | (0.99 to 1.13) | 0.109 |
| **Neurological diseases other than dementia** | | | | | | | | | | |
| Parkinson disease | 274 | (6.7) | 26,575 | (0.2) | 43.94 | (38.85 to 49.70) | <0.001 | 26.21 | (23.15 to 29.68) | <0.001 |
| Epilepsy | 371 | (9.1) | 93,330 | (0.6) | 17.31 | (15.56 to 19.27) | <0.001 | 16.06 | (14.43 to 17.89) | <0.001 |
| Multiple sclerosis | 98 | (2.4) | 49,170 | (0.3) | 8.11 | (6.63 to 9.91) | <0.001 | 8.99 | (7.35 to 10.97) | <0.001 |
| **Psychiatric disorders and related drug therapies** | | | | | | | | | | |
| Substance abuse disorders (drugs, alcohol, and cannabis) | 594 | (14.6) | 154,415 | (1.0) | 17.76 | (16.28 to 19.38) | <0.001 | 18.34 | (16.79 to 20.04) | <0.001 |
| Schizophrenia and psychotic disorders | 328 | (8.1) | 204,721 | (1.3) | 6.85 | (6.11 to 7.67) | <0.001 | 7.12 | (6.36 to 7.97) | <0.001 |
| **Traumatic brain injury** | 259 | (6.4) | 66,133 | (0.4) | 16.58 | (14.61 to 18.81) | <0.001 | 14.02 | (12.35 to 15.92) | <0.001 |

[a] Overall French population without YOD include individuals aged 40 to 64 years old and affiliated to the general security scheme in 2016.

[b] ORs, 95% CIs, and p-values were computed using logistic regression.

[c] OR adjusted for age and sex.

CI, confidence interval; OR, odds ratio; YOD, young onset dementia.

**Table 3. Percentage reduction of the association between male sex and YOD incidence due to comorbidities.**

| Model | OR men versus women[a] | 95% CI | Percentage reduction (%)[b] |
|---|---|---|---|
| **Model 1 adjusted for age** | 1.48 | (1.39 to 1.57) | — |
| **Model 1 + covariates** | | | |
| Morbid obesity | 1.48 | (1.39 to 1.58) | 0.0 |
| Diabetes | 1.46 | (1.37 to 1.55) | 3.5 |
| Antihypertensive drug therapy | 1.48 | (1.39 to 1.57) | 0.0 |
| Acute cerebrovascular disease (excluding transient attacks) | 1.45 | (1.36 to 1.53) | 5.2 |
| Sequelae of cerebrovascular disease or history of acute cerebrovascular disease | 1.41 | (1.32 to 1.50) | 12.4 |
| Acute ischemic heart disease | 1.48 | (1.39 to 1.57) | 0.0 |
| Chronic ischemic heart disease or history of acute ischemic heart disease | 1.44 | (1.36 to 1.53) | 7.0 |
| Cardiac arrhythmias and conduction disorders | 1.44 | (1.36 to 1.53) | 7.0 |
| **All cardiovascular, cerebrovascular, and metabolic disease and related drug therapies** | **1.35** | **(1.27 to 1.43)** | **23.5** |
| Parkinson disease | 1.46 | (1.37 to 1.55) | 3.5 |
| Epilepsy | 1.43 | (1.34 to 1.52) | 8.8 |
| Multiple sclerosis | 1.51 | (1.42 to 1.60) | −5.1 |
| **All neurological diseases other than dementia** | **1.44** | **(1.35 to 1.53)** | **7.0** |
| Substance abuse disorders (drugs, alcohol, and cannabis) | 1.32 | (1.24 to 1.40) | 29.2 |
| Schizophrenia and psychotic disorders | 1.46 | (1.37 to 1.55) | 3.5 |
| **All psychiatric disorders and related drug therapies** | **1.31** | **(1.23 to 1.39)** | **31.1** |
| **Traumatic brain injury** | **1.41** | **(1.33 to 1.50)** | **12.4** |
| **All comorbidities** | **1.19** | **(1.11 to 1.27)** | **55.6** |

[a] ORs, 95% CIs, and p-values were computed using logistic regression. All ORs are significant at $p < 0.001$.

[b] Percentage reduction = $100 \times (logOR_{Model\ 1} - logOR_{Model\ 1\ +\ covariates})/logOR_{Model\ 1}$.

CI, confidence interval; OR, odds ratio; YOD, young onset dementia.

YOD in 2016. Rates of YOD were higher in men compared to women, and we identified that this association was attenuated by 56% after adjustment for all comorbid conditions considered, suggesting a potential role for comorbid conditions in the relationship between sex and YOD.

## Burden of YOD in France and comparison with other studies

Previous field studies have provided inconsistent results regarding YOD prevalence, with PR ranging from 90 to 420/100,000 between 30 and 64 years [11,14,39,40]. Our estimation is compatible with the only previous French study [20], although it reported a slightly lower prevalence in people aged 41 to 60 years (PR = 41.2/100,000, n = 39 versus PR = 67.7/100,000 in our study for the same age group) probably due to the inclusion of young onset Alzheimer disease only, not all-cause dementia. Our results are also in line with 2 studies: one performed in the province of Modena in Italy (2006 to 2019, PR = 119.9/100,000) [41] and the other in 4 catchment areas of eastern Sydney, Australia (2007 to 2008, PR = 133/100,000 between 45 and 64 years; the PR for this age group in our study was 127/100,000) [11]. Regarding incidence, 2 recent field studies provided IR that were consistent with our findings; in the Italian study mentioned above, the IR was 22.8/100,000, and it was 25.0/100,000 between 45 and 64 years in a population-based study in Norway (2015 to 2017) [11,17].

In registry-based studies, PR ranged from 81 to 143/100,000 between 45 and 65 years, in line with our study findings [9,12,13,15,16,40–42]. In addition, 4 registries provided IR. Our estimate (24.4/100,000 person-years) is close to that from a Spanish study (2007 to 2009) that

estimated IR to be 22.8/100,000 person-years between 40 and 65 years [16]. Other studies conducted in Scotland (1974 to 1988) [18], England (2000 to 2006) [19], and Japan (2018 to 2019) [43] yielded much lower IR of 2.7, 11.5, and 2.5/100,000 person-years, respectively. The inclusion of psychiatric hospitals records only in the Scottish study, the small catchment area and thus small number of incident cases in the English study, and a large population a denominator (18 to 64 years) in the Japanese study are the main explanations for the low incidence estimates in these studies.

Other studies used national or regional administrative databases [44–46], with important differences regarding study design, data sources, and criteria used to define YOD. Recently, the European medical information framework consortium analyzed 6 electronic health databases from 5 countries to estimate dementia PR and IR [45]. Our results are comparable to those from 2 databases identifying dementia using general pharmacy and hospital records: 1 from Tuscany (Italy, 50 to 70 years: PR = 100/100,000; IR = 30/100,000 person-years) and 1 from the northern and central regions of Jutland (Denmark, 50 to 70 years: PR = 300/100,000; IR = 58/100,000 person-years). In our study, the PR between 50 and 65 years was 100/100,000, and the IR was 37/100,000 person-years. Most studies that included primary care data provided estimates similar to our own [44,45], but one reported higher rates [46]; it ascertained dementia using 4 linked administrative health databases from the Saskatchewan Province (Canada), a wide range of ICD–10 codes, and cognitive testing data, but not functional tests, which may have led to an overestimation of the YOD frequency.

Given the heterogeneity of the information available in administrative databases in each country, comparisons of YOD frequency are difficult, and differences should be interpreted with caution. A recent review on the accuracy of dementia diagnosis in routinely collected datasets reported good specificity and positive predictive value but moderate sensitivity [24]. Based on 12 studies, sensitivity was comprised between 21% and 86%, with only 3 of 12 studies reporting estimates ≥60%. However, some studies found sensitivity to be higher at younger ages [25,26]. There are 2 major reasons for reduced sensitivity in administrative databases: first, 1 to 3 persons with dementia is undiagnosed and therefore undetected in administrative databases [23]; second, criteria used to identify dementia lack sensitivity since not all persons with dementia are treated with anti-Alzheimer drugs or hospitalized [21,27]. Regarding younger individuals with YOD, cognitive decline and dementia are often associated with behavioral disturbances [4], and they face a more aggressive course with a steeper rate of cognitive decline [47]. Therefore, the disease has a major impact on individuals' lives, which likely explains higher sensitivity of administrative databases in younger individuals who are more likely to seek medical care and integrate the health system than their older counterparts.

## Sex differences in YOD incidence and role of comorbidities

Many studies examined sex differences in dementia, and their results remain equivocal [48,49]. Previous studies mostly reported no differences between men and women for LOD incidence [50,51]. Studies with higher rates in women mostly found differences restricted to the oldest age group [52,53]. For YOD, we found that the $PRR_{M/W}$ was 1.33 and the $IRR_{M/W}$ 1.39 and that this difference decreased with age. Two Japanese multicenter population-based studies (n = 2,469 [42] and n = 4,077 [39] YOD) reported higher YOD prevalence in men than women ($PRR_{M/W}$ = 1.59, 95% CI = 1.56 to 1.53 and mean $PRR_{M/W}$ = 1.4, respectively). Also consistent with our findings, in both the hospital-based Italian and Danish studies discussed above, $PRR_{M/W}$/$IRR_{M/W}$ <70 years were 1.11/1.25 in the Italian (n = 2,659 YOD) and 1.45/ 1.25 in the Danish study (n = 2,183 YOD) [45] and decreased thereafter. Other studies found no significant sex differences for prevalence [11,12,16,40,46] or incidence [11,16,46,50].

Differences between men and women in dementia rates are likely to arise both from differences related to sex (i.e., biological) and gender (i.e., behaviors, such as addictions; sociocultural, such as education or occupation) [49]. A number of reasons may thus account for higher YOD rates in men. First, there are important sex differences in middle age for several comorbidities associated with dementia. The incidence of cardio- and cerebrovascular disease is higher in middle-aged men than women [54]. There is increasing evidence that these conditions are associated with cognitive decline and LOD [31,33]. Although additional studies are needed to better understand their role in YOD, in our study, cardio- and cerebrovascular diseases or metabolic disease explained approximately 23% of the association between male sex and YOD, thus suggesting an important contribution of vascular dementia to YOD. In addition, other disorders associated with cognitive decline and dementia (e.g., substance abuse [55], Parkinson disease [29], epilepsy [35], and traumatic brain injury [56]) are more common in men than women and explained an important part of the association between male sex and YOD. Second, women may benefit from neuroprotective effects of estrogens regarding cerebrovascular disease and dementia [57]. Third, compared to men, women may have a reduced or delayed use of healthcare services for memory complaints. According to the French national Alzheimer Database that includes information for all individuals who attend memory clinics, in 2012, women were older than men (77.5 versus 74.4 years) and had a lower Mini-Mental State Examination (MMSE) score (20.5 versus 21.6 points) at the time of first visit [58]. However, these findings may reflect the higher proportion of older women in France rather than a delayed used of medical care services. Unfortunately, YOD data were not available in this study.

## Strengths and limitations of the study

The main strengths of our study are its large size, the inclusion of all persons aged 40 to 64 years old affiliated to the general scheme of the French social security, and the ascertainment of all persons with dementia who required medical care within that population. The large number of persons with YOD allowed us to estimate PR, IR, and sex ratios with precise CIs. The comparison with a reference cohort of persons without YOD yielded a sex difference similar to the incidence analysis and provides insights about the role of comorbidities in sex differences in YOD incidence.

The main limitation of our study is that we were unable to identify undiagnosed persons and those who were not in contact with the medical system. As discussed above, this is less likely to occur in this age group than in older individuals. In addition, we used the first date of contact for dementia with the medical system to define age at incidence, which could be delayed relative to the actual age at incidence, leading to misclassification of some individuals with YOD as having LOD [59,60]. No data are available in France on the delay between dementia diagnosis and anti-Alzheimer drugs initiation, LTD registration, or first hospitalization. However, French health authorities recommend to provide LTD registration for dementia at the time of diagnosis [61]; in addition, at the time of this study, anti-Alzheimer drugs were fully reimbursed for individuals with LTD registration, while only 15% of their cost were reimbursed to people without registration. These recommendations are likely to lead to a short delay between diagnosis and LTD registration and treatment. Another important limitation pertains to the lack of accurate information regarding dementia subtypes; one of the reasons may be variability in how ICD-10 codes are attributed in different hospitals and by different doctors all over France. Among the 3 criteria used to identify dementia, LTD registrations and hospitalizations have associated ICD-10 codes that could be used for this purpose. However, the accuracy of these codes and their relevance to identify dementia subtypes remain unclear,

and additional validation studies are needed before they can be used for this aim. In addition, the use of anti-Alzheimer medications to identify YOD could have shifted our selection toward the inclusion of a higher number of persons with Alzheimer disease compared to other dementia subtypes. However, the majority of persons with YOD was identified through LTD registration or hospitalizations and only 8% by anti-Alzheimer medications alone. In addition, in France, some of the anti-Alzheimer medications can also be prescribed for other causes of dementia, such as Parkinson disease, dementia with Lewy bodies, or mixed dementia. Regarding study sample selection (76% of the French population), the participants excluded from our study are very heterogeneous. Public services concentrate a higher number of women and well-educated individuals, while agricultural and self-employed workers are 2 groups constituted of more men and with a large heterogeneity in terms of educational level. Therefore, and given the low amount of individuals that are missing, it is unlikely that the selection lead a systematic bias in our analysis. Other limitations include the lack of information on cognitive status and dementia severity and risk or protective factors (e.g., education level, occupation, smoking, alcohol drinking, physical activity, and diet). Finally, drug claims are not available in SNDS for persons in nursing home with an internal pharmacy, but this has little influence on our results given that these institutions are only available in France for persons ≥60 years.

## Policy and research implications

We provide French YOD frequency estimates that help increase awareness on its burden. In addition, we show the important contribution of cardio- and cerebrovascular, metabolic, neurological (other than dementia), psychiatric diseases, and traumatic brain injury in explaining sex differences in YOD incidence. The higher YOD frequency in men deserves further investigation to clarify the role of sex-associated risk factors and comorbidities as well as differences in healthcare access. Our results may suggest that preventive approaches targeting cardiovascular risk factors in midlife, substance abuse disorders, and prevention and management of traumatic brain injury could be further investigated as strategies for reducing or postponing YOD.

## Supporting information

**S1 STROBE Checklist. Checklist of items that should be included in reports of cross-sectional studies.** STROBE, Strengthening the Reporting of Observational Studies in Epidemiology.
(DOCX)

**S1 Table. Algorithms for the identification of comorbidities and treatments of interest in the SNDS.** SNDS, Système National des Données de Santé.
(DOCX)

**S2 Table. PRs (per 100,000 persons) of YOD in France on December 31, 2016.** PR, prevalence rate; YOD, young onset dementia.
(DOCX)

**S3 Table. IRs (per 100,000 person-years) of YOD in France in 2016.** IR, incidence rate; YOD, young onset dementia.
(DOCX)

**S4 Table. Association between comorbidities and sex in persons without YOD from the French population in 2016 (40 to 64 years).** YOD, young onset dementia.
(DOCX)

## Author Contributions

**Conceptualization:** Laure Carcaillon-Bentata, Alexis Elbaz.

**Formal analysis:** Cécile Quintin, Marjorie Boussac-Zarebska.

**Methodology:** Laure Carcaillon-Bentata, Cécile Quintin, Marjorie Boussac-Zarebska, Alexis Elbaz.

**Supervision:** Laure Carcaillon-Bentata, Alexis Elbaz.

**Validation:** Laure Carcaillon-Bentata, Alexis Elbaz.

**Writing – original draft:** Laure Carcaillon-Bentata, Alexis Elbaz.

**Writing – review & editing:** Laure Carcaillon-Bentata, Cécile Quintin, Marjorie Boussac-Zarebska, Alexis Elbaz.

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
