## [Editor Report · Decision Letter 0]

11 Feb 2021

Dear Dr Carcaillon-Bentata, 

Thank you for submitting your manuscript entitled "Early-onset dementia in France: sex-differences in prevalence, incidence, and role of comorbidities" for consideration by PLOS Medicine.

Your manuscript has now been evaluated by the PLOS Medicine editorial staff as well as by an academic editor with relevant expertise and I am writing to let you know that we would like to send your submission out for external peer review.

Kind regards,

Caitlin Moyer, Ph.D.

Associate Editor

PLOS Medicine

---

## [Decision Letter · Decision Letter 1]

2 Jun 2021

Dear Dr. Carcaillon-Bentata,

Thank you very much for submitting your manuscript "Early-onset dementia in France: sex-differences in prevalence, incidence, and role of comorbidities" (PMEDICINE-D-21-00532R1) for consideration at PLOS Medicine. 

Your paper was evaluated by a senior editor and discussed among all the editors here. It was also discussed with an academic editor with relevant expertise, and sent to four independent reviewers, including a statistical reviewer. The reviews are appended at the bottom of this email and any accompanying reviewer attachments can be seen via the link below:

[LINK]

In light of these reviews, I am afraid that we will not be able to accept the manuscript for publication in the journal in its current form, but we would like to consider a revised version that addresses the reviewers' and editors' comments. Obviously we cannot make any decision about publication until we have seen the revised manuscript and your response, and we plan to seek re-review by one or more of the reviewers. 

We expect to receive your revised manuscript by Jun 23 2021 11:59PM. Please email us (plosmedicine@plos.org) if you have any questions or concerns.

We look forward to receiving your revised manuscript. 

Sincerely,

Caitlin Moyer, Ph.D.

Associate Editor 

PLOS Medicine

plosmedicine.org

. Title: Please revise your title according to PLOS Medicine's style. Your title must be nondeclarative and not a question. It should begin with main concept if possible. "Effect of" should be used only if causality can be inferred, i.e., for an RCT. Please place the study design ("A randomized controlled trial," "A retrospective study," "A modelling study," etc.) in the subtitle (ie, after a colon).

2. Data Availability Statement: The Data Availability Statement (DAS) requires revision. For each data source used in your study:

3. Line numbers: Please provide line numbers throughout the manuscript with the revised version.

4. Author Summary: Please move the Author Summary so that it follows the Abstract. Please format using bullet points, with 2-3 points for each section. This text is subject to editorial change and should be distinct from the scientific abstract. Please see our author guidelines for more information: https://journals.plos.org/plosmedicine/s/revising-your-manuscript#loc-author-summary

5. Abstract: Methods and Findings: Please provide some more detail on the study population. Please quantify the main results (with 95% CIs and p values) and please include the important dependent variables that are adjusted for in the analyses. * In the last sentence of the Abstract Methods and Findings section, please describe the main limitation(s) of the study's methodology.

6. Abstract: Conclusions: * Please address the study implications without overreaching what can be concluded from the data; the phrase "In this study, we observed ..." may be useful.

7. In-text citations: Please use numbers within square brackets for in-text citations. Where more than one reference is indicated, please do not include spaces within brackets.

8. Methods: Please ensure that the study is reported according to the STROBE guideline, and include the completed STROBE checklist as Supporting Information. Please add the following statement, or similar, to the Methods: "This study is reported as per the Strengthening the Reporting of Observational Studies in Epidemiology (STROBE) guideline (S1 Checklist)."

9. Methods: Prospective Analysis Plan: Did your study have a prospective protocol or analysis plan? Please state this (either way) early in the Methods section.

10. Results: Please provide p values and 95% confidence intervals for all statistical comparisons, and please present both unadjusted and adjusted results (at least in tables, if not in the text.) Where p values are presented, please clarify the statistical tests used.

11. Discussion: First sentence: Please avoid claims of primacy, or if necessary, please temper with “to the best of our knowledge” or similar.

12. Discussion: If relevant for sex/gender differences, please consider discussing head trauma/ injury, which could be associated with motor vehicle accidents or sports related and may be more prominent in men, and this seems like it could potentially contribute to differences as a risk factor for cognitive decline and dementia.

13. Discussion: Please present and organize the Discussion as follows: a short, clear summary of the article's findings; what the study adds to existing research and where and why the results may differ from previous research; strengths and limitations of the study; implications and next steps for research, clinical practice, and/or public policy; one-paragraph conclusion.

14. References: Please use the "Vancouver" style for reference formatting, and see our website for other reference guidelines https://journals.plos.org/plosmedicine/s/submission-guidelines#loc-references

15. Table 2 and Table 3: Please provide both adjusted and unadjusted analyses.

16. Supplementary Table 1: Please define all abbreviations in the legend.

17. Supplementary Table 4: Please provide both the adjusted and unadjusted analyses.

Comments from the reviewers:

Reviewer #1: Using the French national health data system, this study attempts to estimate EOD prevalence and incidence in 2016 of French individuals aged 40-64y and study the contribution of comorbidities to EOD incidence.

Comments:

2016 is five years ago now. Can the analysis be updated so it is more relevant to todays prevalence and incidence rates?

"Based on the LOD literature, we selected cardio- and cerebro-vascular diseases and treatments, metabolic conditions, psychiatric diseases, psychotropic drugs, and other neurological diseases that may be EOD risk factors."

Could the authors please provide citations for reference here?

The application of Poisson regression to estimate 95% confidence intervals (CI), and male-to-female prevalence (PRRM/W) and incidence (IRRM/W) rate ratios and 95% CI, overall (adjusted for age) and by age, as well as including an interaction term between age and sex, is a technically appropriate and robust methodological approach.

"The IRRM/W was 1.61 (95% CI=1.23-2.11) between 40-44 years and it progressively decreased to 1.41 (95% CI=1.29-1.54) between 60-64 years. "

These results show overlapping CIs, that suggest that in fact the difference is not statistically significant? Did the authors consider undertaking any trend analyses?

Can the authors please present the uncertainty (i.e. the confidence intervals, as shown in supplementary tables 2 and 3) in the PR and IR's in Figure 1?

Can the authors please complete statistical testing of differences between groups in Table 1?

Table 2: Can the authors please include parentheses for the percentage figures?

Reviewer #2: The manuscript of Carcaillon-Bentata et al provides EOD prevalence and incidence estimates in France and the contribution of comorbidities to EOD incidence. To this aim, the authors used the French national health data system. The results they found were they were around 24.000 EOD cases in France in 2016. Further, cardio and cerebrovascular, neurological, and psychiatric disease prevalence explained more than 50% of the sex difference in EOD incidence. The authors highlight that this result supports the idea of preventive approaches aimed at reducing or postponing the onset of dementia by acting on cardiovascular risk factors in mid-life. 

This manuscript has strengths that should be noted. First, the use of the French national health data system which covers more than 98% of the French population. Second, the well-written manuscript. However, I have some doubts about the relevance and the interpretation of the results. 

My first main concern is regarding the interpretation of the findings. The authors have only explored the incidence of EOD encompassing the different types of dementia. It is well-known that the incidence of different types of dementia differs. Due to the higher sample, I would strongly suggest doing a breakdown of the prevalence of different dementia subgroups showing the results in a table with the frequency of clinical diagnosis among EAD cases and incidence rate per 100.000 person-years. Knowing the difference between subgroups would allow for a more powerful discussion of the findings. Also, the impact of comorbidities depends on the type of dementia. Furthermore, previous literature has shown the sex differences in some kinds of dementia, such as Alzheimer's Disease (Ferretti, et al. 2020) and Behavioural Variant of Frontotemporal Dementia (Illán-Gala, et al. 2021). However, the sex difference in the incidence of subtypes of EOD is lacking and deserves further study. Therefore, authors should consider adding this analysis to the current manuscript. 

Following the sex difference topic, in the current manuscript is not clear if the analysis of sex differences in EOD is one of the main aims or a secondary one. In my opinion, I would say that it is one of the main aims because in the methods part the authors specify that "we want to assess the contribution of comorbidities to sex-differences in EAD incidence". If this is correct, I was wondering why they analyzed it, have the authors a previous hypothesis? The manuscript should be rewritten focusing on this topic and giving it more prominence. For instance, in the introduction, there is no information about sex differences in EOD found in the previous literature. Besides, sex differences should be included as a keyword. However, I would strongly encourage the authors to keep this focused on the current manuscript. Elucidating sex differences in dementia, also in its incidence, will allow to development of a precision medicine approach encompassing sex-sensitive strategies for prevention and detection. 

It is well-known the difference between the concepts of sex and gender, referring to biology and social construct respectively. In this line, I was wondering if the gender factors can have an impact on the incidence of EAD as well. 

Reviewer #3: This paper addresses an important topic. Good prevalence and incidence rates for young onset dementia have been lacking and it is important to know these for purposes of policy, service provision and research. This paper therefore makes a welcome contribution to the literature. The report is clear and succinct. The discussion provides a very good comparison with others' findings and the conclusions make a valuable contribution to knowledge in this area. 

I have only four points for consideration:

1. In terms of your data source, I notice you accessed an impressive level (76% of the population) but this excludes those in public service. I wondered whether or not there is any systematic bias here that you could comment upon. 

2. You did not refer to other comparable studies and their methods, strengths and limitations in the introduction, as might be conventional, but left this to the discussion. I think it would help to set the context to refer to these, at least briefly, in the introduction. 

3. I am not a statistician so cannot comment on details of the analysis; however, as a lay person in this respect, I understood the expression, except for the correction used in calculating the incidence rate. This will be only my ignorance but, if shared with other readers, it may be worth considering how to make this more readily understandable. 

4. At the very end of the discussion section, you state that institutionalisation is [only?] available to those >60 years. This seems strange - may I check that this actually correct?

I have two small suggestions in relation to writing style/terminology. 

* Terms used to describe young onset dementia vary but 'young onset dementia' has become the most common, superseding 'early onset dementia', since the latter can become confused with 'early-stage dementia'. I would suggest that you change your terminology to young onset dementia to ensure best fit with emerging nomenclature.

* People living with dementia are very keen to be recognised and respected as whole human beings, especially as the condition is often equated with loss of personhood. In line with this, I would strongly urge you to consider replacing the term 'patient', where possible, with alternatives, e.g. in the first line of the abstract, you could say 'Dementia onset in those aged < 65 years….' And in the first paragraph of the background, you could say '…as the number of people affected….'. This helps to shift attitudes.

Reviewer #4: This a well written, much needed study on the epidemiology of early onset dementia (EOD), a condition with dramatic consequences for the individual patients, their families and the society. It will be important to raise awareness about this condition and for improving planning and allocation of health and social resources. I strongly recommend its publication but I would advise to address the following comments to further improve it: 

1) It should made clear since the beginning, i.e. in the title and in the abstract, the fact that the epidemiological data on EOD where established from a register covering about 76% of the population of France. The current title and abstract give the uncorrect impression that the authors analysed data from the entire population of France. 

2) One major problem of the manuscript in its current version is the scarcity of references to previous epidemiological studies on early onset dementia, some of which are very recent (see as an example Chiari et al 2020). Sentences such as "Few studies have provided precise frequency estimates of EOD" (abstract) and "Some EOD frequency estimates come from registries but they are probably biased due to the inclusion of small numbers of selected patients referred to specialized centers" (introduction) need to be either played down or become more specific about what is actually not precise or lacking in recent work on the epidemiology EOD. 

3) The authors identified dementia cases among all individuals aged 40-64 years old. However, EOD can also occur before the age of 40. It would be very helpful if the authors could extend their analysis to include people aged 30-64. This will be more informative while also allowing a comparison with previous registry-based studies on the epidemiology of EOD (such as Garre-Olmo et al 2010). Please do specify in the first sentence of the Discussion that PR and IR are given for population aged 40-64 (again, I strongly suggest that you also add data regarding the 30-64 age group)

4) One of the strategies for cases identification was the presence of reimbursements over a one-year period of anti-dementia drugs. Please note that acetylcholinesterase inhibitors and memantine are not generically anti-dementia but are specifically licensed for Alzheimer's disease. Although Alzheimer's disease is the most frequent cause of dementia, there are several other diseases causing EOD that especially present in young patients, such as frontotemporal dementia. Please change 'anti-dementia' to a more specific terminology and mention in the discussion that the inclusion of such a strategy may have biased case identification towards AD, possibly missing other dementias that do not have specific medications. 

5) In the discussion, the sentence "Field studies provided inconsistent results regarding EOD" needs to be re-framed. Actually, it may be argued that this sentence is outdated, as the only reference provided is a review paper of 2014, which includes studies published before 2014. Please add and discuss recently published original studies.

Similarly, the sentence "these estimates were generally based on few cases and imprecise" should be reframed. Please only consider stating that others' work is imprecise unless you have clear evidence and can prove with exact numbers such a strong judgment.

6) In the discussion please clearly consider that the lack of information about the type/etiology of dementia is a major limitation of the study, as this may be a much important limitation than the other limitations listed in the same sentence or at least equally important to the lack of information regarding dementia severity. Related to this please mention that the study is likely to be biased towards Alzheimer's disease, given that it is the only type of dementia that has specific medications (as for point 4)

Minor comments:

The sentence "After 60 years, 6-9% of all dementia cases occurred before 65 years" in the first paragraph of the introduction is not easily readable. Doo the authors mean "Among the people older than 60,. …"

What is the reference for the study mentioned in "one from Tuscany (Italy, 50-70 years: PR=100/100,000; IR=30/100,000 person-years)"?

Re the sentence "important contribution of cardio-and cerebro-vascular, metabolic, neurological, and psychiatric diseases in explaining sex-differences in EOD incidence" in the conclusion: since dementia is a neurological disorder, please specify "other-than dementia neurological disorders"

[LINK]

---

## [Decision Letter · Decision Letter 2]

18 Aug 2021

Dear Dr. Carcaillon-Bentata,

Thank you very much for re-submitting your manuscript "Young onset dementia: prevalence, incidence, and role of comorbidities using French administrative databases" (PMEDICINE-D-21-00532R2) for review by PLOS Medicine.

I have discussed the paper with my colleagues and the academic editor and it was also seen again by four reviewers. 

Provided the remaining editorial and production issues are dealt with we are planning to accept the paper for publication in the journal.

However, there are a number of remaining editorial issues that need to be addressed are listed at the end of this email. Any accompanying reviewer attachments can be seen via the link below. Please take these into account before resubmitting your manuscript:

[LINK]

We look forward to receiving the revised manuscript by Aug 25 2021 11:59PM.   

Sincerely,

Caitlin Moyer, Ph.D.

Associate Editor 

PLOS Medicine

plosmedicine.org

Requests from Editors:

1. Title: Reviewer 3 makes a specific suggestion for the title; however, we suggest: “Prevalence and incidence of young-onset dementia and associations with comorbidities: A study of data from the French national health data system” or similar.

2. Data availability statement: The statement requires revision, as it currently reads that data are found in the supporting information files, which is not the case. Please include information explaining how individuals may request data access, such as: We suggest: “Data cannot be shared as access to the SNDS has to be approved by the French Data Protection Authority. Interested investigators can request their own access to the data according to the French law. Information on obtaining access to SNDS data can be found at https://www.snds.gouv.fr/SNDS/Processus-d-acces-aux-donnees. Requests for access to NSDS data must be submitted to the National Institute for Health Data (INDS)[[please provide a link or contact email]].”

3. Throughout: Please replace "subject" with participant, patient, individual, or person.

4. Abstract: Methods and findings: Please include the number of individuals that are included among the 76% of the French population in the French National Health Data System, and please specify the name (Système national des données de santé (SNDS)). Please provide p values for the comparison of PR and IR between men and women. Please provide the result (OR with 95% CI and p value) for the association between specific comorbidities mentioned and incident YOD. 

5. Abstract: Conclusions:Line 45-46: Please revise to “...higher YOD frequency in men may be partly explained by higher prevalence…”

6. Author summary: Why was this study done?: We suggest changing the second point to: “To the best of our knowledge, there is a lack of data regarding the frequency of young onset dementia…”

7. Author summary: What do these findings mean?: Please revise the final point to: “YOD incidence was higher in men than women, and this could possibly be due, at least in part, to the higher prevalence of…” or similar.

8. In-text citations: Please place the reference bracket before the sentence punctuation, rather than after.

9. Introduction: Line 74: We suggest noting here that young-onset dementia can also be referred to as early-onset dementia.

10. Introduction: Line 101: Here and throughout, please replace the term “subjects” with “individuals” or similar.

11. Methods: Ethical approval: Please state in the Methods that the relevant institutional review board(s) waived the need for ethical approval for the study.

12. Methods: Analysis plan: Please make sure that the Methods section transparently describes when the described analyses were planned, and when/why any data-driven changes to analyses took place. Changes in the analysis-- including those made in response to peer review comments-- should be identified as such in the Methods section of the paper, with rationale. For example, please note whether the prevalence, incidence and comorbidities analyses were preplanned, and please note that the analysis of comorbidities in the context of sex differences was planned after observing differences in YOD between sexes. Please also note that the analysis of TBI was also added after the main analyses were completed.

13. Methods: Line 111: We suggest “Data on hospitalizations in short stay departments....and at home are available.”

14. Methods: Line 134, Line 153 and Line 159: Please replace “subjects” with “individuals” in these sentences.

15. Methods: Line 143-151: Please provide more details, or a list in the Supporting Information files, of the specific comorbidities assessed, unless this is described in supplementary table S1.

16. Methods: Line 168-169: Please provide a reference to the National Institute of Statistics and Economic Studies data used here.

17. Methods: Line 180-181: Please include the reference to the STROBE Checklist in the Supporting Information file (S1 Checklist, or similar).

18. Results: Line 187-188: Please revise this sentence to clarify: “Women (12.7%) more often met all three criteria for inclusion compared to men (7.9%)...” or similar.

19. Results: Line 209: Please report p values as <0.001 where applicable.

20. Results: Line 219: Please clarify if “delusional disorders” is meant, or if psychotic disorder was intended.

21. Discussion: Line 227: We suggest removing the word “robust” from the sentence.

22. Discussion: Line 231-232: We suggest revising to: “Rates of YOD were higher in men compared to women, and we identified that this association was attenuated by 55% after adjustment for all comorbid conditions considered, suggesting a potential role for comorbid conditions in the relationship between sex and YOD.” or similar.

23. Discussion: Line 241: Please change to “...four catchment areas of eastern Sydney, Australia…”

24. Discussion: Line 264: Please change “overestimate” to “overestimation” if accurate.

25. Discussion: Line 267-270: Please revise this sentence to clarify, as the meaning is unclear: “A recent review on the accuracy of dementia diagnosis in routinely collected datasets reported good specificity and positive predictive value with moderate sensitivity,[24] comprised between 21%-86%, with only 3 of 12 studies reporting estimates ≥60%; however, sensitivity was higher at younger ages.[25, 26].”

26. Discussion: Line 277: Please replace the term “subject” with “individuals” or similar.

27. Discussion: Line 313: Please clarify “inclusions of all persons <65 years” as your study seemed to consider adults less than 65 yrs of age but at least 40 years.

28. Discussion: Line 322: Please replace “subject” with “individual” or similar.

29. Discussion: Line 324: Please replace “subject” with “individual” or similar. We suggest clarifying “overestimate” in this sentence by rephrasing “...which could be delayed relative to the actual age at incidence, leading to misclassification of some individuals with YOD as having LOD.” or similar.

30. Discussion: Line 343: Limitations: Please comment on whether it is possible that people with Parkinson’s disease but not dementia are included in your study because they are receiving donepezil, for example?

31. Discussion: Line 344 and Line 348: Please replace the term “subject” with “individual” or similar.

32. Discussion: Line 360: We suggest revising to “Our results may suggest that preventive approaches targeting cardiovascular risk factors in mid-life, substance-abuse disorders, and prevention and managing traumatic brain injury could be further investigated as strategies for reducing or postponing young onset of dementia.” or similar, to make it clear that these results cannot demonstrate causal relationships between comorbidities and young onset dementia.

33. Sections “Disclosure of potential conflicts of interest” “Funding” and “Availability of data and material” in the main manuscript text: Please remove these from the manuscript body, and please make sure all information is accurately entered with the manuscript submission metadata. 

34. References: Please check the formatting of each reference in the list. Please use the "Vancouver" style for reference formatting, and see our website for other reference guidelines https://journals.plos.org/plosmedicine/s/submission-guidelines#loc-references

Specifically, please check the NLM abbreviations of journal titles; for example, “The Lancet Neurology” should be “Lancet Neurol” in references 1 and 2. Please also check capitalization (e.g. “Bmj” in reference 18).

35. Table 1 and Figure 1: Please define the abbreviation YOD in the legend.

36. Figure 1: Please clarify that the shaded region represents the 95% CIs.

37. Figure 2: Please clarify that the bars represent the 95% CIs. Please define IRR and PRR in the legend.

38. Table 1: Please report p values as <.001 where relevant rather than <.0001.

39. Table 2 and Table 3: We suggest replacing “psychiatric diseases” with “disorders” and “delusional diseases” with “delusional disorders” in the tables.

40. Table 2: In the footnote, please clarify the source of the comparison group “Overall French Population without YOD (2016). Please also report p values as <.001 where applicable.

41. Supplementary table 1: We suggest “psychiatric disorders” instead of “diseases” in the table. We suggest changing “delusional diseases” to “psychotic disorders” or similar, as the ICD-10 codes seem to include disorders related to psychosis, in addition to including delusional disorders. In the legend, please briefly explain “t” and “t-4” as used in the study.

42. Supplementary table 2: Please define YOD in the legend. Please clarify “(Rniam)” in the legend. Here and for supplementary table 3, please provide a reference for the number of persons/ person-years affiliated to the general scheme of the social security in 2016.

43. Supplementary table 4: We suggest “psychiatric disorders” instead of “diseases” in the table. We suggest changing “delusional diseases” to “psychotic disorders” or similar, as the ICD-10 codes seem to include disorders related to psychosis, in addition to including delusional disorders. Please also report p values as <.001 where applicable.

44. STROBE Checklist: We suggest renaming the file as S1 Checklist or S1 Strobe Checklist, or similar.

Comments from Reviewers:

Reviewer #1: The authors have responded to each comment in turn. 

Can the authors please re-visit the following conclusion, and change the tone from 'likely' to 'possibly' due to insufficient statistical power?

"Age-specific IRRM/W were similar to corresponding age-specific PRRM/W, and there was a similar trend of decreasing IRRM/W with continuous age but it was not statistically significant (p-interaction=0.494),

likely due to insufficient statistical power as the number of incident cases in each age group was considerably smaller than the number of prevalent cases."

Reviewer #2: The authors have sold all my comments addressing them clearly and correctly. 

Reviewer #3: Thank you for sending a revision of this paper and for giving such a clear account of changes made. I have four minor points arising and have also listed a equally small number of spelling/grammatical changes to address. 

1. I noted the new title but would like to suggest streamlining to:

"Young onset dementia in France: prevalence, incidence and role of comorbidities."

2. The editors asked that, in the abstract, you should quantify the main results (with 95% CIs and p values) - please check you have done this.

3. I also noted the request for you to include the (completed) STROBE checklist. I did not see the checklist in the supplementary materials, despite a heading on p.45.

4. Lines 246-251. Can you hypothesise the reason for the lower rates in then Scottish, English and Japanese studies?

Errors of spelling or grammar:

1. Line 112: "After each hospital stay, a discharge summary is filled." - was this intended to say 'filed'?

2. Line 234 - add two words: 'Previous Field studies have provided inconsistent results regarding YOD prevalence

3. Line 264: which may have led to overestimation of YOD frequency.

4. Lines 267-270 sentence beginning 'A recent review on the accuracy of dementia diagnosis …' would be better broken into two.

5. Line 309: 'However, these findings ….'

Reviewer #4: I have read the response to mine and to the other Reviews with great interest and appreciation. All my comments have been addressed and my questions answered properly. I strongly recommend publication.

[LINK]

---

## [Editor Report · Decision Letter 3]

8 Sep 2021

Dear Dr Carcaillon-Bentata, 

On behalf of my colleagues and the Academic Editor, Perminder Singh Sachdev, I am pleased to inform you that we have agreed to publish your manuscript "Prevalence and incidence of young-onset dementia and associations with comorbidities: A study of data from the French national health data system" (PMEDICINE-D-21-00532R3) in PLOS Medicine.

Please also address the following editorial requests:

-Where multiple references are included within brackets, please do not include spaces. For example, at line 86, this should be [6,7].

-Results: Line 231 and Line 235: Please also include the p values for these results in the text (it seems they would both be p<0.001).

PRESS

Sincerely, 

Caitlin Moyer, Ph.D. 

Associate Editor 

PLOS Medicine